# Generalizability of Deep Learning Models for Caries Detection in Near-Infrared Light Transillumination Images

**DOI:** 10.3390/jcm10050961

**Published:** 2021-03-01

**Authors:** Agnes Holtkamp, Karim Elhennawy, José E. Cejudo Grano de Oro, Joachim Krois, Sebastian Paris, Falk Schwendicke

**Affiliations:** 1Department of Oral Diagnostics, Digital Health and Health Services Research, Charité-Universitätsmedizin Berlin, 14197 Berlin, Germany; agnes.holtkamp@charite.de (A.H.); jose-eduardo.cejudo@charite.de (J.E.C.G.d.O.); Joachim.krois@charite.de (J.K.); 2Department of Operative and Preventive Dentistry, Charité-Universitätsmedizin Berlin, 14197 Berlin, Germany; sebastian.paris@charite.de; 3Department of Orthodontics, Dentofacial Orthopedics and Pedodontics, Charité-Universitätsmedizin Berlin, 14197 Berlin, Germany; karim.elhennawy@charite.de

**Keywords:** artificial intelligence, caries, diagnostics, digital imaging/radiology, mathematical modeling

## Abstract

Objectives: The present study aimed to train deep convolutional neural networks (CNNs) to detect caries lesions on Near-Infrared Light Transillumination (NILT) imagery obtained either in vitro or in vivo and to assess the models’ generalizability. Methods: In vitro, 226 extracted posterior permanent human teeth were mounted in a diagnostic model in a dummy head. Then, NILT images were generated (DIAGNOcam, KaVo, Biberach), and images were segmented tooth-wise. In vivo, 1319 teeth from 56 patients were obtained and segmented similarly. Proximal caries lesions were annotated pixel-wise by three experienced dentists, reviewed by a fourth dentist, and then transformed into binary labels. We trained ResNet classification models on both in vivo and in vitro datasets and used 10-fold cross-validation for estimating the performance and generalizability of the models. We used GradCAM to increase explainability. Results: The tooth-level prevalence of caries lesions was 41% in vitro and 49% in vivo, respectively. Models trained and tested on in vivo data performed significantly better (mean ± SD accuracy: 0.78 ± 0.04) than those trained and tested on in vitro data (accuracy: 0.64 ± 0.15; *p* < 0.05). When tested in vitro, the models trained in vivo showed significantly lower accuracy (0.70 ± 0.01; *p* < 0.01). Similarly, when tested in vivo, models trained in vitro showed significantly lower accuracy (0.61 ± 0.04; *p* < 0.05). In both cases, this was due to decreases in sensitivity (by −27% for models trained in vivo and −10% for models trained in vitro). Conclusions: Using in vitro setups for generating NILT imagery and training CNNs comes with low accuracy and generalizability. Clinical significance: Studies employing in vitro imagery for developing deep learning models should be critically appraised for their generalizability. Applicable deep learning models for assessing NILT imagery should be trained on in vivo data.

## 1. Introduction

For detecting and assessing dental caries lesions, visual-tactile and radiographic methods, often in combination with each other, have been the standard over recent decades. Over the last years, Near-Infrared Light Transillumination (NILT), an alternative to radiography for caries lesion detection and assessment has been developed and tested, for example, DIAGNOcam (KaVo, Biberach, Germany). NILT makes it possible to assess teeth in motion (thereby providing some three-dimensional assessment) or to record videos or images. In contrast to radiography, no ionizing radiation is generated. Further, the device is portable and can be repeatedly applied in children and in short intervals, for example, in high-risk individuals. NILT has been confirmed to show similar accuracies for detecting proximal caries lesions to radiography by both in vitro and in vivo studies [1,2,3,4,5,6], and a recent meta-analysis demonstrated the underlying evidence to be robust [7].

For assessing diagnostic imagery such as radiographs or NILT, individual examiners rely on their training and experience. A wealth of studies have shown that dentists show limited accuracy (often being associated with their experience) and inter- and intra-examiner reliability when assessing diagnostic images [5,6,7,8]. Automated assistance systems based on deep learning, for example, using Convolutional Neural Networks (CNNs), might help to overcome these limitations, reducing false detections and diagnostic and treatment variability, especially of less experienced dentists. CNNs are machine learning models consisting of stacked layers of linear models, corresponding weights, and a bias term and nonlinear functions. They differentiate from other neural networks using convolution operations to extract features, particularly from imagery, such as edges, corners, spots, and macroscopic patterns. As with all machine learning models, they learn data from pairs of datapoints, in this case, images and labels (e.g., caries lesion present yes/no, or the areas or pixels affected by a caries lesion). By repeatedly passing these data-label pairs through the network and optimizing the model weights during training, CNNs are able to identify the inherent statistical patterns and to eventually predict a label on unseen data [9]. In dentistry, CNNs have been applied to detect caries lesions, periodontal bone loss, and apical lesions on radiographs [10].

In two previous publications, it was shown that CNNs can also be used for caries detection on NILT imagery attained in vitro [11] or in vivo [12]. Imagery obtained in vitro can come with a hard ground truth (as teeth can be sectioned and histologically evaluated after the assessment via NILT) but inherent biases routed in their acquisition (extracted teeth mounted in simulation models), while in vivo imagery usually results in a fuzzy ground truth (based on the labeling by multiple human experts) but reflects the clinical situation more realistically. From a researcher’s point of view, it could be useful to train a CNN on clinical data and test it on in vitro data (where a firm ground truth can be obtained). Clinicians, on the other hand, would like to know if CNNs developed in vitro are also applicable clinically. The present study aimed to train CNNs on NILT imagery attained in vitro and in vivo and aimed to test their generalizability.

## 2. Materials and Methods

### 2.1. Study Design

This study employed 2 datasets of NILT imagery, 1 generated in vitro for a previous study [11] and the other generated in vivo. Images were pixel-wise annotated for proximal caries lesions by 3 independent dental specialists followed by a master reviewer. The segmentation masks were then transformed into binary labels (caries present yes/no). ResNet-type deep CNNs for binary classification were trained on a training dataset from 1 source and tested on hold-out datasets from both sources. Reporting of this study follows the STARD guideline [13] and the Checklist for Artificial Intelligence in Medical Imaging, CLAIM [14].

### 2.2. Performance Metrics

We used k-fold cross-validation with 10 train, validation, and test splits for evaluating the performance of the models, which were optimized using binary cross-entropy as the loss function. For measuring performance, 6 metrics were employed: F1-score, sensitivity, specificity, predicted positive value (PPV), negative predictive value (NPV,) and area under the receiver operating characteristic curve (AUC).

### 2.3. Sample Size

No formal sample size calculation was performed. For both datasets, a comprehensive sample of available imagery was employed.

### 2.4. Dataset

For the in vitro dataset, 226 extracted posterior teeth (113 premolars and 113 molars) were obtained with informed consent under an ethics-approved protocol (ethics committee of Charité, EA4/102/14), as described before [11]. The teeth were embedded in transparent epoxy resin (Epo-Thin 2, Buehler, Lake Bluff, IL, USA), which allowed the resulting NILT imagery to be similar to clinically attained images for the human eye. The models were mounted in a dummy head (Phantomkopf P-6, Frasaco, Tettnang, Germany). NILT images were acquired using the DIAGNOcam by 1 examiner (KE), moving the camera perpendicularly to the occlusal surface over each tooth. The dental unit light was switched off during this examination. Images were captured using the KID software (KaVo Integrated Desktop/version 2.4.1.6821, KaVo), with each image focusing on 1 tooth.

For the in vivo dataset, 1319 images from routine examination data at Charité-Universitätsmedizin Berlin, obtained since 2019, were used (as approved by Charité ethics EA4/080/18) A comprehensive sample of patients aged 18 years or older receiving NILT, radiographic, and visual-tactile examination over a maximum time period of 12 months was included, resulting in 56 patients. NILT was employed by 1 experienced dentist (AH).

### 2.5. Reference Test

The reference test constituted the pixel-wise annotations of proximal primary caries lesions by 3 independent and experienced dentists (clinical experience: 8–11 years), followed by the review of all annotated images by 1 master reviewer who was able to evaluate and revise all existing annotations. The union of all pixel labels on each image remaining after the review was obtained, and the final classification reference set was established by transforming the segmentation masks into binary targets (caries present yes/no). The decision to first provide pixel-wise annotations was chosen to allow segmentation modeling at some point. For this study on generalizability, a classification model was used as outlined. Each annotator independently assessed each image under standardized conditions using an in-house custom-built annotation tool as described before [11]. Examiners were informed about the study and all trained annotations on a separate set of 10–50 NILT images. Note that we did not score the lesions into further classes such as incipient or advanced lesions. Although this would have been possible based on the pixel-based annotations, it was not of interest within the classification task of this study.

### 2.6. Data Preparation, Model, and Training 

A Residual Convolutional Neural Network (ResNet) was used for classification. The model takes an RGB image as an input and outputs probabilities for binary classes corresponding to sound and carious teeth. The architecture of our residual network was a stack of residual blocks of CNNs as a feature extractor and a fully connected classification head with binary output. We used the feature extraction blocks from a pretrained model on Imagenet of the Pytorch library.

For augmentation, random rotations, vertical and horizontal flipping, shifting, and zooming were applied during training, with a probability of 0.5. The images were resized to 224 × 224 × 3 tensors, which proved to be enough resolution for this task. 

The performance of the model on the in vivo and in vitro datasets was assessed separately using k-fold cross-validation with 10 train, validation, and test splits, respectively. When the training and evaluation data were from different sources (e.g., training on in vivo and testing on in vitro data), all the images from the training source were used for training the model. The generalizability of the model was then assessed by splitting the data from the testing dataset into validation and test splits. The validation splits were evaluated during training, and the test splits were evaluated after the models had converged. This process was repeated for 10 different splits. 

For each split, we trained for 200 epochs using the Adam optimizer. The binary cross-entropy loss on the validation data was monitored during training with a patience parameter of 20 epochs, after which early stopping was applied and the mean and standard deviation of the outlined performance metrics were evaluated across the test sets.

Due to the small size of the in vitro data set, it was challenging to set a proper combination of batch size and learning rate that yielded a stable behavior when training with this dataset. Learning rates of 5 × 10^−7^, 5 × 10^−6^, 5 × 10^−5^, and 5 × 10^−4^ and batch sizes of 4, 8,16, 32, and 64 were considered, respectively. We performed a grid search for each possible combination. The best combination can be found in the Appendix A. Our models were trained on a NVIDIA Quadro RTX 6000 graphics card (NVIDIA, Santa Clara, CA, USA).

### 2.7. Explainability

Another aim was to make the models interpretable. There are several visualization algorithms for deep learning models available, and we chose to use GradCAM [15]. This algorithm creates a visualization that makes it possible to distinguish the most salient areas relevant for a particular class. This saliency map is computed as a weighted combination of the feature maps of a particular layer followed by a ReLU activation. The coefficients of the combination are computed as the average of the gradients of the output class with respect to the feature maps. 

### 2.8. Statistical Analysis

Differences in model performance were evaluated via independent 2-sided t-tests, using *p* < 0.05 as a discriminating criterion. Computations were performed using the Python library SciPy 1.5.2.

## 3. Results

The tooth-level prevalence of caries lesions was 41% in vitro and 49% in vivo, respectively. The models trained on in vivo data performed significantly better (mean ± SD accuracy and AUC: 0.78 ± 0.04) than those trained in vitro when tested on the imagery from the same dataset (accuracy: 0.64 ± 0.15, AUC: 0.65 ± 0.12, *p* < 0.05). Within each analysis, sensitivity and specificity were similar, while the NPV was significantly higher than the PPV.

When tested in vitro, the models trained in vivo showed significantly lower accuracy (0.70 ± 0.01) and AUC (0.66 ± 0.01) (*p* < 0.01). Similarly, when tested in vivo, models trained in vitro showed significantly lower accuracy (0.61 ± 0.04) and AUC (0.60 ± 0.04) (*p* < 0.05). In both cases, this was due to decreases in sensitivity (by −27% for models trained in vivo and −10% for models trained in vitro). Specificity values did not change or even increased slightly when cross tested. NPV values decreased by −13% and −6%, while decreases in PPV were only −5% and −3%. Models trained in vitro and tested in vivo no longer showed useful sensitivity and PPV. Figure 1 sums up the ROC curves for all four scenarios.

When assessing salient areas found relevant by the models to come to a decision (Figure 2), it became apparent that for true positive detections, the most relevant pixels were also those that dentists found to be affected by caries. False-positive detections on in vivo imagery were often associated with restorations, with a similar appearance to carious lesions. In vitro, it was not always possible to identify reasons behind the models’ decision. For false-negative detections, it became clear that the models found other areas than the lesion relevant, indicating an attention problem.

## 4. Discussion

NILT is an imaging method for caries detection and assessment. As the device is portable and near-infrared light not being ionizing, it offers a range of advantages over radiography. It specifically lends itself for usage in outreach settings such as schools, care homes, or nondental clinics, for example, in the hands of dental auxiliary or nondental staff. Notably, assessing NILT imagery is challenging, with limited accuracy and low inter-examiner reliability. Using deep learning via CNNs may support NILT diagnostics. There are currently two studies available on CNNs for NILT diagnostics, one conducted in vitro-generated image material and one on in vivo-generated image material [11,12]. In vivo, it is hard to establish a solid ground truth, as histologic assessment or other means (microradiography, µCT) are not available [16], while in vitro, such validation is possible, but sufficiently large sample sizes (e.g., thousands of tooth segments) for training and testing of CNNs are hard to attain. Hence, this study aimed to evaluate models trained in vivo and tested in vitro and vice versa showed generalizability. The research question is relevant beyond specific use cases and has not been explored in dentistry so far.

Based on the findings, the generalizability of models trained on imagery from one source is not necessarily given when tested on imagery from another source. Generally, models trained on in vitro data showed limited accuracy, despite prevalence rates being similar. Notably, the dataset available for training was much smaller in vitro than in vivo, and accuracy (and possibly generalizability) can be expected to increase if larger datasets are used. Overall, in vitro trained models were at the border of being useful or, when applied on in vivo data, no longer useful. In vivo models also showed a drop in accuracy when tested for their generalizability but remained useful.

Interestingly, when tested on the same data material, both models showed similar sensitivities and specificities, but when tested on the other image source, sensitivity dropped drastically while specificity remained stable. Obviously, the learned pattern to identify carious lesions on one image material was not readily applicable on the other dataset. This was confirmed by visualization. In case of false-negative detections, models focused completely on other areas than those marked as carious by dentists. For false-positive detections, we found the presence of fillings to affect models’ decisions on in vivo material, while no such pattern emerged in vitro. It should be highlighted that pixels relevant for true-positive detections were similar to those areas marked by dentists as constituting carious lesions.

This study has a number of strengths and limitations. First, it is one of few generalizability studies in the field of deep learning and, more so, deep learning in dentistry. Second, the trained models showed useful accuracies, at least when trained on in vivo imagery. Third, employing methods of explainable AI allowed inference toward the models’ decision-making, strengthened confidence into the results, and allowed insights as to reasons behind false classifications. Fourth, as a limitation, the sample sizes, especially in vitro, were limited. Moreover, the in vitro ground truth was not established via histology but by the independent marking of affected pixels and review by experts. This process is obviously not without bias but has been previously used in a range of studies [10]. Fifth, the prevalence of carious lesions was rather high in both datasets compared with other reports on proximal carious lesions (e.g., a study from Sweden found 1.3 proximal lesions in molars or premolars, translating to 1.3 per 20 proximal surfaces being carious, or 6.5%) [17]. Whereas having a balanced dataset helps training CNNs, any PPVs or NPVs will be biased by spectrum bias and should be interpreted accordingly. Sixth, we only used tooth segments for training, making no use of clustering effects (and associated correlation structures) or further context [18]. Last, this study did not evaluate how using a CNN to assist NILT diagnostics impacts clinical care and decision making. This was an active decision, as our focus was a methodological one, and future studies should explore this in detail.

## 5. Conclusions

Using in vitro setups for generating NILT imagery and training CNNs comes with low accuracy and generalizability. Models trained and tested in vivo showed higher accuracy but limited generalizability when tested on in vitro data. Studies employing in vitro image materials for developing deep learning models should be critically appraised for their generalizability. Applicable deep learning models for assessing NILT imagery should be trained on in vivo data.

## Figures and Tables

**Figure 1 jcm-10-00961-f001:**
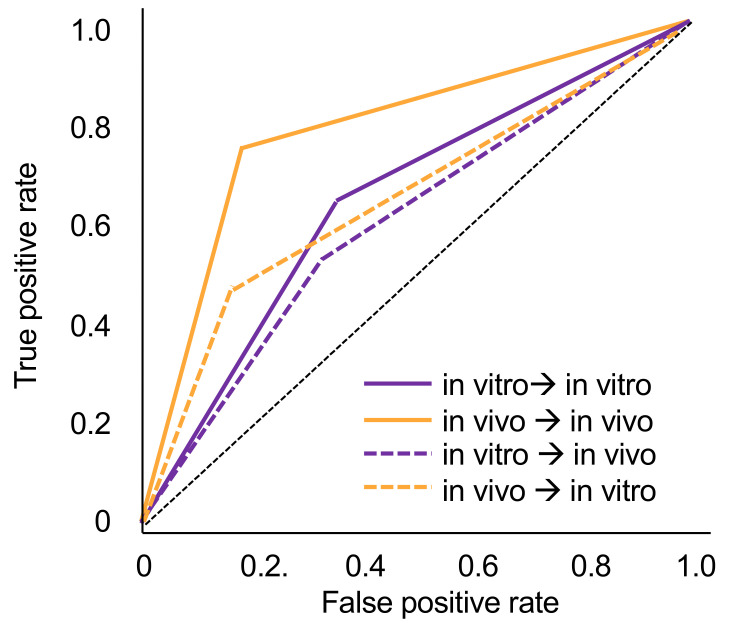
Receiver operating characteristic (ROC) curves for models trained in vivo and in vitro, respectively, and tested on data from the same or the other data source. The resulting area under the curve (AUC) values can be found in Table 1.

**Figure 2 jcm-10-00961-f002:**
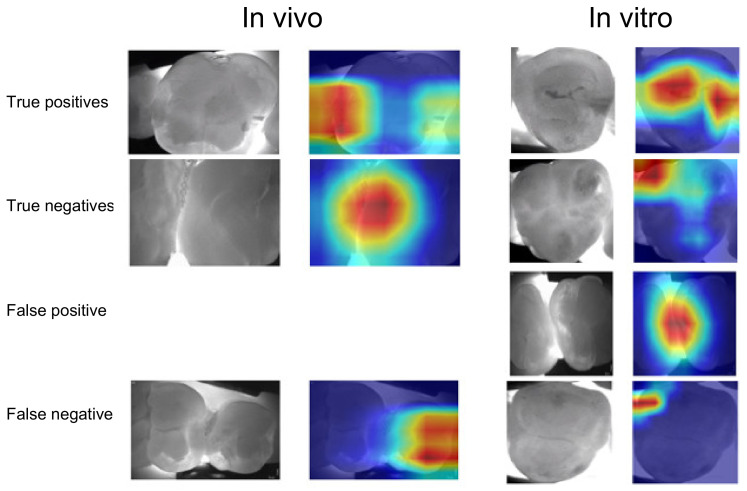
Visualization of contributing feature maps. The original images and the salient areas the models found most relevant for their decision (highlighted in yellow to red) are shown. Data for models trained in vivo and in vitro, respectively, and tested on data from the same or the other data source are shown.

**Table 1 jcm-10-00961-t001:** Mean performance ± standard deviation of the models trained in vivo and in vitro, respectively, and tested on data from the same or the other data source.

Trained → Tested	Accuracy	F1-Score	AUC	Sensitivity	Specificity	PPV	NPV
in vitro	0.64 ± 0.15	0.57 ± 0.16	0.65 ± 0.12	0.66 ± 0.12	0.64 ± 0.21	0.55 ± 0.22	0.76 ± 0.09
in vivo	0.78 ± 0.04	0.73 ± 0.04	0.78 ± 0.04	0.76 ± 0.06	0.79 ± 0.05	0.70 ± 0.05	0.84 ± 0.03
trained in vitro → tested in vivo	0.61 ± 0.04	0.52 ± 0.03	0.60 ± 0.04	0.55 ± 0.03	0.65 ± 0.07	0.49 ± 0.05	0.70 ± 0.03
trained in vivo → tested in vitro	0.70 ± 0.01	0.56 ± 0.03	0.66 ± 0.01	0.49 ± 0.06	0.83 ± 0.04	0.67 ± 0.04	0.71 ± 0.03

AUC: Area under the curve. PPV/NPV: Positive/negative predictive value.

## Data Availability

Data cannot be made available given data protection reasons.

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
