# Peer review of "Generalizability of Deep Learning Models for Caries Detection in Near-Infrared Light Transillumination Images"

_jcm, 2021, doi:10.3390/jcm10050961_

Round 1
Reviewer 1 Report
The manuscript addresses a topic that is of current interest and evaluates the current caries detection/assessment. It would be informative to include the diagnosis of the caries, such as incipient, primary, recurrent caries, etc if possible. This will clarify the accuracy and limitations of detecting caries from clinical perspectives.
Author Response
We added the terminology of lesions suggested to place our study more within the clinical arena.
Reviewer 2 Report
This study aimed to evaluate in vitro and in vivo trained CNNs on NILT images and to analyze whether sufficient accuracy and generalizability is given when applied to the respective opposite data sets. The topic is of current interest and the results are relevant. Therefore, the study should be accepted after a minor revision.
The following aspects should be considered for improvement:
Instead of the first person plural "We aimed", a more neutral writing style in the third person singular should be preferred: "This study aimed...".
Spaces before and after mathematical symbols should be added.
Line 137 f: Duplication, of the aim. The reader is already informed about it.
Line 146: 10 >> ten
Line 150: Description of the descriptive statistics, can be omitted or should be stated under 2.8.
Line 154: The learning rate numbers are difficult to read. Commas between the values and spaces before/after mathematical signs, as well as the formula sign × instead of the letter x are preferable.
Line 157: ...(company, headquarters).
Line 172 ff: Space before and after mathematical signs. Please add the unit.
Line 175: Even if the abbreviations NPV and PPV are familiar, they should be written out in full in the text at the first time.
Table 1: The label "train vitro test vivo, train vivo test vitro" should be improved in terms of wording.
Line 247: second >> fifth?
Line 253 fifth > sixth
Refrences
Verify Ref 3
Author Response
A more neutral writing style was used.
Spaces before and after mathematical symbols were added.
Duplication of the aim was removed.
Line 146: 10 >> ten
Description of the descriptive statistics was omitted.
The learning rate numbers, including the usage of commas between the values and spaces before/after mathematical signs, as well as the formula sign × instead of the letter x were changed.
Units were added.
The abbreviations NPV and PPV were given.
Table 1: The label "train vitro test vivo, train vivo test vitro" was altered.
Minor changes as suggested were made.
Reference 3 was corrected.